# Hard Samples, Bad Labels: Robust Loss Functions That Know When to Back Off

## Abstract

Incorrectly labelled training data are frustratingly ubiquitous in both benchmark and specially curated datasets. Such mislabelling clearly adversely affects the performance and generalizability of models trained through supervised learning on the associated datasets. Frameworks for detecting label errors typically require well-trained / well-generalized models; however, at the same time most frameworks rely on training these models on corrupt data, which clearly has the effect of reducing model generalizability and subsequent effectiveness in error detection — unless a training scheme robust to label errors is employed. We propose two novel loss functions, *Blurry Loss* and *Piecewise-zero Loss*, that enhance robustness to label errors by de-weighting or disregarding difficult-to-classify samples, which are likely to be erroneous. These loss functions leverage the idea that mislabelled examples typically appear as outliers to their as-labelled class, being difficult to classify, and should contribute less to the learning signal. Comprehensive experiments on a variety of both artificially corrupted and real-world datasets demonstrate that the proposed loss functions outperform state-of-the-art robust loss functions in nearly all cases, achieving superior F1 and Balanced Accuracy scores for error detection. Further analyses through ablation studies offer insights to confirm the mechanism through which these loss functions operate, and demonstrate their broad applicability to cases of both uniform and non-uniform corruption, and with different label error detection frameworks. By using these robust loss functions, machine learning practitioners can more effectively identify, prune, or correct errors in their training data. Code, including a working demonstration Jupyter Notebook, is available at `https://anonymous.4open.science/r/Robust_Loss-6BAD/`.

## 1 Introduction

Supervised learning relies heavily on the assumption that training labels are accurate; however, label errors are pervasive even in well-established benchmark datasets, typically at rates of at least 5% (Northcutt et al., 2021b), including in classic datasets such as MNIST (Deng, 2012; Northcutt et al., 2021b). Label errors (also referred to as label *noise* in the literature) degrade model performance, limit generalizability (Song et al., 2022; Zhang et al., 2016; Pleiss et al., 2020), and mislead validation metrics. The prevalence of label errors is especially problematic in domains with hierarchical and fine-grained classes, such as biological datasets (Wu et al., 2019; Garcin et al., 2021; Van Horn et al., 2021; He et al., 2024; Nguyen et al., 2024; Gharaee et al., 2024a;b), where mislabelling often occurs among highly similar categories. Efficient automatic error detection could significantly streamline curation and the flagging of samples for expert review, reducing costs and improving dataset quality.

Existing work on detecting label errors (Jiang et al., 2018; Song et al., 2019; Kim et al., 2019; Pleiss et al., 2020; Northcutt et al., 2021a) such as Confident Learning (CL) (Northcutt et al., 2021a) and Area Under Margin (AUM) (Pleiss et al., 2020) typically rely on training surrogate models, intended to be robust to overfitting, such that they are well-generalized and not fit to erroneous labels in the training data. Crucially, the effectiveness of these methods depends on the models producing statistically distinguishable predicted probabilities, $p(k|x)$, for erroneously *vs.* correctly labelled samples. However, when trained with standard loss functions such as Cross Entropy (Good, 1952) or Focal Loss (Lin, 2017), models tend to inadvertently fit to

erroneous labels, reducing detection effectiveness. Modification of the loss function used when training such models is one avenue for improving their robustness and downstream utility for detecting label errors.

In this paper, we introduce and rigorously investigate two novel robust loss functions — *Blurry Loss* and *Piecewise-zero Loss* — specifically designed to improve model robustness by explicitly de-weighting samples likely to be mislabelled. In datasets with label errors, erroneous samples appear out of the distribution of their as-labelled class and are likely to be difficult-to-classify. Inspired by Focal loss (Lin, 2017), which places emphasis on difficult-to-classify samples, these novel loss functions reverse this idea, assigning less weight to and reversing or zeroing gradients for samples with low predicted probability in their as-labelled class (likely to be mislabelled), thereby avoiding detrimental overfitting to erroneous data. We illustrate our approach in Figure 1. These loss functions result in a more robust training scheme and models that are better suited to detecting label errors than those trained with existing standard or robust loss functions.

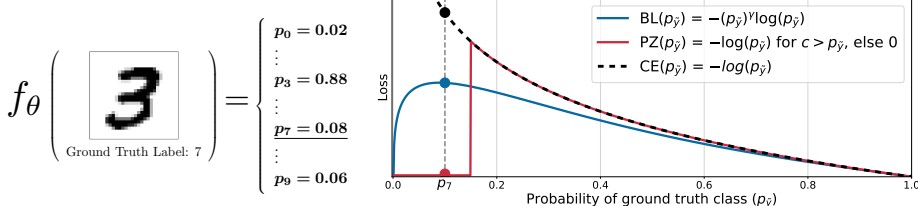

Figure 1: A model being trained operates on a corrupted data sample with a label error (image of "3" labelled as 7), and yields a low predicted probability for the ground truth class as labelled, $p_7$. Cross Entropy or Focal Loss would be sensitive to such samples, steering the model to fit to erroneous data. In contrast, the proposed loss functions, Blurry Loss (BL) and Piecewise-zero Loss (PZ), are insensitive and lead to robust training.

Comprehensive experiments are performed to validate these loss functions' effectiveness in improving label error detection, exploring several real-world label error datasets as well as artificially corrupted datasets, a range of artificial corruption rates, uniform and non-uniform error statistics, multiple frameworks for label error detection (CL and AUM), and gradients and predicted probability distributions of corrupted and clean training samples. In particular, the results of Figures 3 and 4, showing trends in gradients and predicted probability distributions during training, demonstrate that the proposed loss functions operate as intended to improve robustness to mislabelled data. The proposed Blurry and Piecewise-zero loss functions outperform existing loss functions, Generalized Cross Entropy (Zhang & Sabuncu, 2018) (GCE) and Active-Negative Loss (Ye et al., 2023) (ANL), across nearly all experiments, highlighting their utility for practical error detection and correction.

## 2 Preliminaries

Building upon Zhang & Sabuncu (2018); Ye et al. (2023), this work is set in the context of a $K$-class classification problem based with $\mathcal{X} \subset \mathbb{R}^r$ as the $r$-dimensional feature space (typically images) and $\mathcal{Y} = \{1, \ldots, K\}$ as the label space (categorical). A clean (ideal, error-free) dataset is denoted by $D = \{(x_i, y_i)\}_{i=1}^{N}$ for sets containing $N$ examples and where $(x_i, y_i) \in (\mathcal{X}, \mathcal{Y})$. A classifier is a function, $f : \mathcal{X} \to \mathcal{Y}$, that maps input features to predicted classes by selecting the class, $k$, having the maximal predicted probability, $f = \arg_k \max p(k|x)$. The predicted probabilities, $p(k|x)$, for each class $k \in \mathcal{Y}$, are inferred through a trainable deep neural network (DNN) with a softmax output layer, such that $\sum_{k=1}^{K} p(k|x) = 1$. In practice, classifiers are trained through the strategy of attempting to minimize the empirical risk, $R_{\mathcal{L}} = \frac{1}{N} \sum_{i=1}^{N} \mathcal{L}(p(k|x_i), y_i)$ for a given loss function $\mathcal{L} : [0, 1]^K \times \mathcal{Y} \to \mathbb{R}^+$.

Label errors are often present within training data, as discussed in Section 1. The proportion of data that are mislabelled is termed the corruption rate, $\eta \in [0, 1]$, and the corrupted dataset is denoted $D_\eta = \{(x_i, \tilde{y}_i)\}_{i=1}^{N}$ for possibly erroneous labels $\tilde{y}_i$. The probability that a correct label $y$ is mislabelled (corrupted) as label $\hat{y}$ is $\eta_{y\hat{y}}$, such that the corruption rate of specific label $y$ is given by $\eta_y = \sum_{\hat{y} \neq y} \eta_{y\hat{y}}$, and the overall corruption rate $\eta$ is a weighted average of $\eta_y$, weighted by the number of examples of each class $N_y$, such

that $\eta = \frac{1}{N} \sum_{\forall y} \eta_y N_y$. Taking the common assumption that the label corruption (*i.e.*, mislabelling) process is conditionally independent of input features (Natarajan et al., 2013), $x$, labels are given by

$$\tilde{y} = \begin{cases} y, & \text{with probability } (1 - \eta_y), \\ \hat{y}, \ \hat{y} \in \mathcal{Y} \setminus \{y\}, & \text{with probability } \eta_{y\hat{y}}. \end{cases} \tag{1}$$

Corruption processes may be categorized based on their nature. In the most general case of *asymmetric* corruption, the corruption process is class-dependent and conditioned on both correct labels, $y$, and corrupted labels, $\hat{y}$. Corruption is *symmetric* if the process is *identically* conditioned on both correct and incorrect labels. Lastly, the corruption process is *uniform* if it is not conditioned on either correct or incorrect labels.

For the sake of notational simplicity throughout the remainder of this article, predicted probabilities, $p(k|x)$, are stated in terms of an *implied* input $x$, such that the notational simplification is made, from $p(k|x)$ to $p_k$ (indicating predicted probability for class $k$).

## 3 Background

The background discussion is divided into two main categories: Label Error Detection, and Robust Loss Functions. Because the label error detection frameworks of Confident Learning (CL) (Northcutt et al., 2021a) and Area Under the Margin (AUM) (Pleiss et al., 2020) are used in this paper to evaluate the proposed loss functions, more detail is given to them than to other frameworks. Similarly, the state-of-the-art robust loss functions, Generalized Cross Entropy (GCE) (Zhang & Sabuncu, 2018) and Active Negative Loss (ANL) (Ye et al., 2023), are used in this paper and are presented in more detail.

### 3.1 Label Error Detection

The prevalence and negative impacts on training models caused by errors in training data have motivated the development of label error detection frameworks. Two prominent approaches in this space are Confident Learning (CL) (Northcutt et al., 2021a) and Area Under the Margin (AUM) (Pleiss et al., 2020), both of which rely on training surrogate models on corrupt datasets in order to detect samples likely to have label errors. Other recent frameworks include MentorNet (Jiang et al., 2018), SELFIE (Song et al., 2019), and SelNLPL (Selective Negative Learning and Positive Learning) (Kim et al., 2019).

Confident Learning (Northcutt et al., 2021a) uses k-fold cross-validation to train models and detect errors on all folds of the mislabelled dataset, wherein, for each fold, the joint distribution of observed, $\tilde{y}$, and underlying true labels, $y$, termed the Confident joint, $Q_{\tilde{y},y} := p(\tilde{y}|y)$, is estimated based on the model's predicted probabilities. Proposal methods for sample pruning available in the framework include Prune by Class (PBC), Prune by Noise Rate (PBNR), and "both". PBC detects examples from each class that the model is least confident about, whereas PBNR detects examples that are most likely to be mislabelled as a *different* class. With PBC, for each class $k$, examples with the lowest self-confidence (*i.e.*, $p(k|x)$, for inputs $x$ and labelled class $k$) are proposed for pruning. With PBNR, examples from each class $k$ having the largest margin, (*i.e.*, $p(j|x) - p(k|x)$, $j \neq k$), are proposed for pruning. If the "both" method is selected, examples must both be proposed by PBC and PBNR to be detected.

The Area Under the Margin (Pleiss et al., 2020) method identifies mislabelled samples based on their training dynamics. AUM measures the difference between the logit values, $z_{\tilde{y}}(x)$, for a training example's as-labelled class, $\tilde{y}$, and its highest other class, $\max_{k \neq \tilde{y}} z_k$, and averages over the course of training. The margin is defined as $M(x, \tilde{y}) = z_{\tilde{y}}(x) - \max_{k \neq \tilde{y}}\{z_k\}$, and the Area Under the Margin is calculated over $T$ training epochs as $\text{AUM}(x, \tilde{y}) = \frac{1}{T} \sum_{t=1}^{T} M_t(x, \tilde{y})$. Correctly labelled samples tend to exhibit positive AUM values, while mislabelled samples tend to have negative AUM values due to conflicting gradient updates from other samples of the same class. To determine at which threshold of the AUM statistic samples should be detected as being likely mislabelled, a small portion of the training data, $D_{\text{Threshold}} \subset D_{\text{Train}}$, are re-labelling to a new *fake* class, numbered $K + 1$ for a $K$-class problem, and correspondingly one additional output neuron is added to the DNN. The AUM for this set of 'threshold samples', *i.e.*, $\text{AUM}(x_i, \hat{y}_i)$ for $(x_i, \hat{y}_i) \in D_{\text{Threshold}}$, is monitored, and the 99th percentile AUM from this set is used as the threshold at which correctly- and mislabelled data are separated.

### 3.2 Robust Loss Functions

The choice of loss function largely determines the training dynamics and the generalization properties of models trained under supervision. For classification problems, Categorical Cross Entropy (CE) (Good, 1952) loss is standard. For predicted probabilities $p_k$ and as-labelled class $\tilde{y}$, Cross Entropy loss is defined as $\text{CE}(p_k, \tilde{y}) = -\log(p_{\tilde{y}})$. Focal loss (FL) (Lin, 2017) builds upon Cross Entropy loss by placing additional weight on difficult-to-classify samples, through the inclusion of an additional factor, $(1 - p_{\tilde{y}})^\gamma$, with weighting parameter $\gamma$ (with a standard setting of $\gamma = 2$). Focal loss is defined as $\text{FL}(p_k, \tilde{y}) = -(1 - p_{\tilde{y}})^\gamma \log(p_{\tilde{y}})$. Note that if $\gamma = 0$, Focal loss reduces simply to Cross Entropy. Focal loss primarily sees use in classification problems where some classes are more difficult to learn than others, for example in highly imbalanced datasets where some classes have far fewer training examples from which to learn. Despite their widespread use, standard loss functions are sensitive to label errors (Ghosh et al., 2017; Patrini et al., 2017; Song et al., 2022) as a result of the strong penalization of confident mispredictions, wherein $p_k$ is high for some $k \neq \tilde{y}$ but $p_{\tilde{y}}$ is low for erroneous $\tilde{y} = \hat{y}$.

Robustness to label errors has motivated several alternative loss functions. Mean Absolute Error (MAE) is theoretically robust to symmetric label corruption (Ghosh et al., 2015; 2017), but has been shown to perform poorly on complicated datasets (Zhang & Sabuncu, 2018). Generalized Cross Entropy (GCE) (Zhang & Sabuncu, 2018) combines Cross Entropy with MAE to obtain the positive characteristics of both losses. The GCE loss, officially termed $\mathcal{L}_q$ loss, is defined as the negative Box-Cox transformation(Box & Cox, 1964),

$$\text{GCE}(p_k, \tilde{y}) = \mathcal{L}_q(p_k, \tilde{y}) = \frac{(1 - p_{\tilde{y}}^q)}{q}, \tag{2}$$

where $q \in (0, 1]$. The parameter $q$ controls the transition between these two losses, such that in the limit as $q \to 0$, GCE becomes Cross Entropy (via L'Hôpital's rule), and for $q = 1$, GCE becomes MAE. The recommended parameter setting is $q = 0.7$ for most problems (Zhang & Sabuncu, 2018).

Active Negative Losses (ANL) (Ye et al., 2023) set the current state-of-the-art and are a class of loss functions that combine Normalized Loss Functions (Ma et al., 2020), $\mathcal{L}_{\text{norm}}$, (originally introduced to be robust to label errors; normalizes by dividing loss for as-labelled class with losses summed over all classes) and Normalized Negative Loss Functions (NNLFs), $\mathcal{L}_{\text{nn}}$, (introduced in Ye et al. (2023)) via weighting parameters, $\alpha, \beta > 0$,

$$\text{ANL}(p_k, \tilde{y}) = \alpha \cdot \mathcal{L}_{\text{norm}} + \beta \cdot \mathcal{L}_{\text{nn}}. \tag{3}$$

Any existing loss function, such as Cross Entropy or Focal loss, can be converted to an ANL through the substitution of normalized and normalized negative forms. Following Ye et al. (2023), we additionally apply L1 regularization to mitigate overfitting, weighted by $\delta$. Importantly, $\delta$ is treated as an additional ANL parameter whose recommended value varies across datasets and ANL variants.

Other notable robust loss functions include Symmetric Cross Entropy (Wang et al., 2019) (combines Reverse Cross Entropy (Pang et al., 2018) with CE), PHuber-CE (Menon et al., 2020) (composite loss-based gradient clipping applied to CE), Active Passive Loss (APL) (Ma et al., 2020) (designed to maximize $p_{\tilde{y}}$ while minimizing $p_k$, $k \neq \tilde{y}$), Asymmetric Loss Functions (ALFs) (Zhou et al., 2021), and Curriculum Loss (Lyu & Tsang, 2019). These precursors set the stage for continued research into robust loss design.

## 4 Method

Two novel loss functions are proposed that are designed to be robust to label errors. During training on corrupt data, $D_\eta = \{(x_i, \tilde{y}_i)\}_{i=1}^N$, mislabelled samples $(x, \hat{y})$ are likely difficult to classify and thus have low predicted probabilities, $p_{\tilde{y}} = p_{\hat{y}}$, for their as-labelled class, $\hat{y}$. With CE loss (Good, 1952), having low predicted probability, $p_{\tilde{y}}$, means the *gradient* of the loss is *large and negative*, imparting a strong signal to the optimizer causing fitting to these samples. The proposed loss functions achieve their robustness by reducing the loss for samples likely to be mislabelled and assign either a *positive* or *zero* gradient, no longer incorrectly steering the optimizer towards fitting to these erroneous samples. This mechanism is experimentally verified in the ablation studies of Figures 3 and 4 in Section 6.2. The proposed loss functions are outlined here, whereas the experimental details are outlined in Section 5.

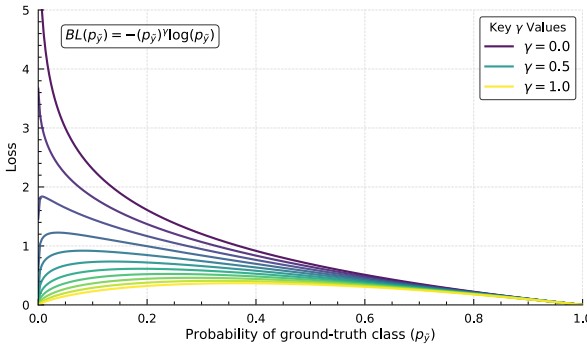 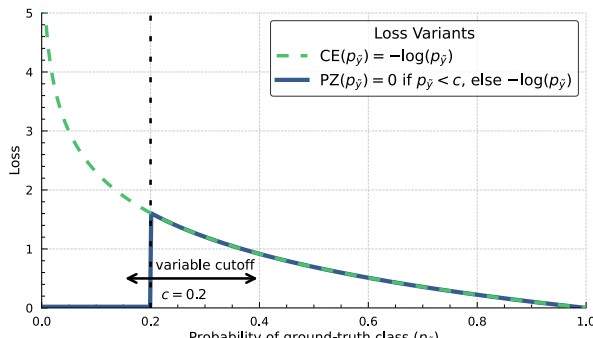

(a) The Blurry Loss function of Equation (4), plotted for a range of parameter $\gamma$, controlling the shape. Note that at $\gamma = 0$, this is equivalent to CE.

(b) The Piecewise-zero Loss function of Equation (5), illustrating the effect of adjusting the cutoff position, $c$. Note that at $c = 0$, this is equivalent to CE.

Figure 2: The two proposed loss functions: Blurry Loss (a) and Piecewise-zero Loss (b).

### 4.1 Blurry Loss

*Blurry Loss* is closely related to Focal loss (Lin, 2017), but *de-weights* difficult-to-classify samples, likely to be mislabelled, through a multiplicative factor. For predicted probability $p_k$, as-labelled class $\tilde{y}$, and a weighting parameter $\gamma \in \mathbb{R}$, Blurry Loss is defined as

$$\mathrm{BL}(p_k, \tilde{y}) = -(p_{\tilde{y}})^\gamma \log(p_{\tilde{y}}). \tag{4}$$

Figure 2a shows the this loss function for several variations of parameter $\gamma$. Notice that this function is non-monotonic and features a section of *positive* gradient for $p_{\tilde{y}} < e^{-1/\gamma}$, encouraging training *against* the as-labelled class if $p_{\tilde{y}}$ is low. At $\gamma = 0$, Blurry Loss is equivalent to Cross Entropy Loss.

### 4.2 Piecewise-zero Loss

*Piecewise-zero Loss* is a variant of CE loss and is designed to ignore (zero) difficult-to-classify samples, which are likely to be mislabelled. Examples with predicted probability beneath some cutoff, $p_{\tilde{y}} \leq c \in [0, 1]$, are assigned a loss of zero through a piecewise definition of the function. Also note that the *gradient* within the cutoff region is also zero, causing these examples with low $p_{\tilde{y}}$ not to affect the training process (weights are not updated by zero-gradient samples). Piecewise-zero Loss is defined as

$$\mathrm{PZ}(p_k, \tilde{y}) = \begin{cases} 0 & p_{\tilde{y}} \leq c, \\ \mathrm{CE}(p_k, \tilde{y}) = -\log(p_{\tilde{y}}) & p_{\tilde{y}} > c. \end{cases} \tag{5}$$

Figure 2b shows this loss function and the action of the cutoff-position parameter $c$. At $c = 0$, Piecewise-zero Loss is equivalent to Cross Entropy Loss.

### 4.3 Loss Scheduling

DNN model parameters are typically randomly initialized at the start of training. As a result, the model's outputs at the early stages of training bear no meaningful relationship to the training labels. Instead, untrained DNNs tend to assign disproportionately high probability to a small subset of classes while assigning nearly zero probability to the majority, nearly regardless of the input provided. This phenomenon is known as Initial Guessing Bias (IGB) (Francazi et al., 2024). As a result of IGB, immediately applying the proposed loss functions — especially the Piecewise-zero Loss with a high cutoff — when predicted probabilities for nearly all classes are nearly zero, is likely to lead to many correctly labelled samples being de-weighted or ignored, dramatically reducing the ability to train. Instead, a scheme through which training begins with Cross Entropy loss is proposed, to allow the model to sufficiently generalize prior to the use of these novel

loss functions. A delay parameter, $d \in \mathbb{N}$, determines the epoch at which the training switches to one of the proposed loss functions. With this scheduling approach, for example with PZ loss, the loss, $\mathcal{L}_{\text{PZ},d}$, used is

$$\mathcal{L}_{\text{PZ},d}(p_k, \tilde{y}) = \begin{cases} \text{CE}(p_k, \tilde{y}) & \text{Epoch} \leq d, \\ \text{PZ}(p_k, \tilde{y}) & \text{Epoch} > d. \end{cases} \tag{6}$$

## 5 Experimental Setup

The following paragraphs outline the datasets, method of artificial corruption, loss functions used for comparison, models and training scheme, and metrics used for the experiments.

**Datasets and Artificial Corruption**  Synthetic uniform label noise is injected into standard benchmark datasets following the same method used in Natarajan et al. (2013); Zhang & Sabuncu (2018); Ye et al. (2023); Pellegrino et al. (2023). For a given corruption rate $\eta \in [0, 1]$, a fraction $\eta$ of the training labels are randomly flipped. Corruption rates in $\eta \in \{0.1, 0.2, 0.3, 0.4\}$ are tested, as this range reflects typical label noise levels encountered in real-world datasets. Datasets include MNIST (Deng, 2012), Fashion-MNIST (Xiao et al., 2017), CIFAR-10, and CIFAR-100 (Krizhevsky & Hinton, 2009), under the assumption that their existing label error rates are negligible compared to the induced corruption, $\eta$.

To assess robustness under realistic, non-uniform noise, the CIFAR-10N and CIFAR-100N (Wei et al., 2022) datasets are employed, which are versions of the CIFAR-10 and CIFAR-100 datasets, re-annotated using Amazon Mechanical Turk. By design, the re-annotations do *not* correct labels but rather induce abundant, real-world, non-uniform human annotation errors. The original CIFAR labels are considered ground truth in the CIFAR-N datasets. For CIFAR-10N, three re-annotations are given, whereas for CIFAR-100, only one is given (with $\eta = 0.4020$). With CIFAR-10N, the 'Aggregate' version (labels used are the majority vote of the three re-annotations; $\eta = 0.0903$) and the 'Worst' version (re-annotations *not* matching ground truth are always used when available; $\eta = 0.4021$) are used for experiments.

In order to evaluate performance as dataset size scales up, we evaluate on two large-scale datasets, Food-101N (Lee et al., 2018) with 310k images and Clothing-1M (Xiao et al., 2015) with 1M images. We train on non-uniform, web-scraped noisy labels in the training sets, which have respective noise rates of $\eta \approx 0.2$ and $\eta \approx 0.4$, and evaluate on test sets which have verified clean labels.

**Baseline Loss Functions**  Categorical Cross-Entropy (Good, 1952) and Focal Loss (Lin, 2017) are the primary baseline loss functions, as these serve as the *de facto* gold standard for most supervised classification problems. Focal Loss is used with $\gamma = 2$ as recommended in Lin (2017). To assess robustness under label errors, state-of-the-art robust loss functions serve as a baseline. These include Generalized Cross-Entropy (Zhang & Sabuncu, 2018) (GCE) loss, defined in Equation (2), Active Negative Cross-Entropy (Ye et al., 2023) (ANL-CE) and Active Negative Focal Loss (Ye et al., 2023) (ANL-FL), defined in Equation (3).

Consistent with the original formulation, the GCE hyperparameter $q$ is fixed at 0.7 as is recommended in Zhang & Sabuncu (2018). For the ANL variants, the dataset-specific hyperparameters $\alpha$, $\beta$, and $\delta$, as well as the focal-loss scaling factor $\gamma$ for ANL-FL, are set exactly as prescribed in Ye et al. (2023). However, since the Fashion-MNIST dataset was not used in Ye et al. (2023), no parameter specification was provided and thus the MNIST parameters of Ye et al. (2023) are used for the Fashion-MNIST experiments here. Each loss function is evaluated using these recommended settings to ensure a fair comparison.

**Models and Hyperparameters**  For MNIST and Fashion-MNIST, a minimal convolutional neural network is adopted, consisting of two convolutional layers followed by two fully connected layers interleaved with dropout, totally approximately 1.2 million trainable parameters. Under a ten-epoch training schedule, this architecture reliably exceeds 99% test accuracy on clean data (near state-of-the-art) (Kadam et al., 2020). For CIFAR-10, CIFAR-100, and their noisy variants (CIFAR-10N/CIFAR-100N), an ImageNet-pretrained ResNet-34 (He et al., 2016), gently modified for lower resolution images (replaced its initial $7 \times 7$, stride-2 convolution with a $3 \times 3$, stride-1 convolution and removed the following max-pool layer, preserving spatial resolution on $32 \times 32$ inputs), is fine-tuned. Training is performed for ten epochs under identical conditions

using the Adam optimizer (Kingma, 2014) with an initial learning rate of $1 \times 10^{-4}$; a scheduler reduces the learning rate by a factor of 0.1 at epoch 5 for CIFAR-10 (factor 0.2 for CIFAR-100). Under this setup, the model attains 93.5% test accuracy on CIFAR-10 and 74.5% on CIFAR-100. All experiments utilize the Adam optimizer, leveraging its adaptive learning capabilities without requiring additional tuning. For Food-101N and Clothing-1M, to account for the larger dataset size and higher-resolution images, we use an ImageNet-pretrained ResNet-50 model and train for 15 epochs.

To explore the parameter-sensitivity of the proposed losses, a grid search is performed over the Blurry Loss exponent, $\gamma$, and the Piecewise-Zero Loss cutoff, $c$, thereby characterizing their influence on label error detection efficacy and allowing optimal values to be identified for each problem. As demonstrated in ablation Figure 11 of Section A.2, introducing a one-epoch delay yields the best performance for PZ loss; accordingly, $d = 1$ is used for all subsequent PZ loss experiments.

**Label Error Detection and Performance Metrics**  To evaluate each loss function's efficacy for the detection of label errors, the Confident Learning (CL) (Northcutt et al., 2021a) and AUM (Pleiss et al., 2020) frameworks, both described in Section 3.1, are applied. The CL pruning method used in all cases is "both", in which detected samples must both be unlikely to be their as-labelled class and likely to be of another different class. Five cross-validation folds are used with 80% train / 20% predict splits. The use of AUM serves as an ablation study of label error detection, to demonstrate the efficacy of the proposed loss functions across multiple detection frameworks. Following the methodology of Pleiss et al. (2020), experiments on CIFAR-100 are replicated, using a ResNet-34 (He et al., 2016) model trained for 150 epochs with a learning rate of 0.1 and weight decay of $10^{-4}$; however, with Cross-Entropy (CE) (Good, 1952) loss replaced with Blurry and Piecewise Zero Loss for a direct comparison.

Performance is primarily quantified using the F1 score (Van Rijsbergen, 1979) — a gold-standard metric defined as the harmonic mean of precision and recall. The F1 score penalizes situations in which either precision or recall fall, which is not always reflective of one's goals when attempting to identify label errors. To address this limitation with low corruption rates, the Balanced Accuracy score (Brodersen et al., 2010), defined as the average of sensitivity (true positive rate) and specificity (true negative rate), which provides a more robust measure of detector performance under class imbalance, is also reported.

# 6 Results

Results for the main experiment (Section 6.1) and additional studies (Section 6.2) are presented here. All tables are formatted such that best scores are **bolded and underlined**, and second best are simply **bolded**.

## 6.1 Main Findings

The proposed loss functions are compared against existing standard and robust loss functions, including CE, FL, GCE, ANL-CE, and ANL-PZ. This comparison is performed on several datasets and corruption rates, as described in Section 5. Results from 20 random trials are summarized in Table 1. The proposed loss functions are evaluated over a range of their parameters, $\gamma$ and $c$; however, only the best performing cases are shown for brevity. In nearly all cases, the use of the proposed BL and PZ losses result in better performance than with the baseline loss functions.

At low corruption rates ($\eta = 0.1$), FL occasionally results in a higher F1 score since it causes CL to detect more conservatively, favouring precision over recall compared to the proposed loss functions. At these low rates, the detection problem is highly imbalanced: high recall may be achieved while making few correct detections, but precision is dramatically reduced with relatively few false detections. In a context where detected examples are reviewed by experts, prioritizing recall over precision can be advantageous, as catching more errors generally outweighs the cost of incorrectly detecting a few more examples. Precision and recall are examined for BL and PZ, and compared to FL in Figure 7 of Section A.2, where it is found that by tuning the parameters of the proposed loss functions, recall is increased significantly, but at the cost of some precision. To capture a more reasonable objective at lower corruption rates, Balanced Accuracy (mean of sensitivity and specificity) is reported in Table 2, in which the proposed BL and PZ perform strongly throughout.

Table 1: Label error detections, averaged over 20 random trials, measured using the F1 score ($\pm\sigma$). The parameters ($\gamma$ and $c$) resulting in the best performance for each dataset and corruption rate are given for the proposed loss functions, Blurry Loss (BL) and Piecewise-zero Loss (PZ).

| Dataset | $\eta$ | CE | FL | GCE | ANL-CE | ANL-FL | BL | $\gamma$ | PZ | $c$ |
|---|---|---|---|---|---|---|---|---|---|---|
| MNIST | 0.10 | 0.969±0.002 | 0.960±0.004 | 0.957±0.002 | 0.941±0.002 | 0.940±0.002 | **0.979±0.002** | 0.3 | **0.974±0.001** | 0.005 |
| | 0.20 | 0.975±0.001 | 0.967±0.002 | 0.975±0.001 | 0.970±0.001 | 0.969±0.001 | **0.984±0.001** | 0.4 | **0.983±0.001** | 0.010 |
| | 0.30 | 0.975±0.001 | 0.967±0.001 | 0.983±0.001 | 0.980±0.001 | 0.980±0.001 | **0.988±0.001** | 0.4 | **0.986±0.001** | 0.020 |
| | 0.40 | 0.973±0.001 | 0.963±0.002 | 0.987±0.001 | 0.985±0.000 | 0.985±0.001 | **0.990±0.000** | 0.5 | **0.988±0.000** | 0.040 |
| Fashion-MNIST | 0.10 | 0.824±0.003 | **0.831±0.004** | 0.737±0.004 | 0.683±0.003 | 0.679±0.004 | 0.826±0.003 | 0.1 | 0.818±0.003 | 0.005 |
| | 0.20 | 0.884±0.002 | 0.879±0.003 | 0.850±0.002 | 0.822±0.002 | 0.819±0.002 | **0.887±0.002** | 0.2 | 0.885±0.003 | 0.005 |
| | 0.30 | 0.907±0.002 | 0.899±0.001 | 0.900±0.001 | 0.881±0.002 | 0.879±0.001 | **0.917±0.001** | 0.4 | 0.916±0.001 | 0.020 |
| | 0.40 | 0.918±0.002 | 0.896±0.033 | 0.927±0.001 | 0.915±0.001 | 0.914±0.002 | **0.934±0.001** | 0.5 | 0.933±0.001 | 0.040 |
| CIFAR-10 | 0.10 | 0.740±0.006 | 0.757±0.006 | 0.773±0.006 | 0.698±0.006 | 0.693±0.007 | 0.787±0.005 | 0.5 | **0.794±0.008** | 0.040 |
| | 0.20 | 0.773±0.004 | 0.789±0.004 | 0.858±0.004 | 0.821±0.004 | 0.817±0.005 | 0.860±0.004 | 0.6 | **0.863±0.004** | 0.060 |
| | 0.30 | 0.771±0.004 | 0.798±0.003 | 0.893±0.003 | 0.873±0.003 | 0.871±0.003 | 0.894±0.003 | 0.7 | **0.894±0.003** | 0.080 |
| | 0.40 | 0.769±0.003 | 0.801±0.003 | 0.909±0.003 | 0.901±0.002 | 0.899±0.003 | 0.912±0.003 | 0.8 | **0.912±0.003** | 0.100 |
| CIFAR-100 | 0.10 | 0.503±0.009 | **0.520±0.009** | 0.485±0.009 | 0.399±0.007 | 0.397±0.007 | 0.507±0.009 | 0.3 | **0.510±0.009** | 0.010 |
| | 0.20 | 0.628±0.006 | 0.638±0.006 | 0.652±0.007 | 0.578±0.006 | 0.575±0.007 | 0.661±0.007 | 0.5 | **0.667±0.006** | 0.020 |
| | 0.30 | 0.680±0.006 | 0.688±0.006 | 0.742±0.006 | 0.682±0.006 | 0.680±0.006 | 0.744±0.006 | 0.6 | **0.748±0.006** | 0.020 |
| | 0.40 | 0.712±0.004 | 0.718±0.003 | 0.800±0.004 | 0.751±0.004 | 0.746±0.004 | 0.800±0.003 | 0.7 | **0.801±0.004** | 0.020 |

Table 2: As in Table 1, but here reporting the Balanced Accuracy metric ($\pm\sigma$). The proposed BL and PZ perform strongly throughout.

| Dataset | $\eta$ | CE | FL | GCE | ANL-CE | ANL-FL | BL | $\gamma$ | PZ | $c$ |
|---|---|---|---|---|---|---|---|---|---|---|
| MNIST | 0.10 | 0.980±0.002 | 0.971±0.004 | 0.992±0.001 | 0.992±0.000 | 0.992±0.000 | **0.993±0.001** | 0.4 | **0.992±0.001** | 0.040 |
| | 0.20 | 0.981±0.001 | 0.974±0.002 | 0.992±0.000 | 0.991±0.000 | 0.991±0.000 | **0.993±0.000** | 0.5 | **0.993±0.000** | 0.020 |
| | 0.30 | 0.980±0.001 | 0.971±0.001 | 0.992±0.000 | 0.991±0.000 | 0.990±0.000 | **0.993±0.000** | 0.5 | **0.993±0.000** | 0.020 |
| | 0.40 | 0.976±0.001 | 0.966±0.002 | 0.991±0.000 | 0.990±0.000 | 0.989±0.000 | **0.992±0.000** | 0.5 | 0.991±0.000 | 0.040 |
| Fashion-MNIST | 0.10 | 0.942±0.002 | 0.941±0.002 | **0.949±0.002** | 0.943±0.001 | 0.942±0.001 | **0.951±0.001** | 0.5 | 0.948±0.002 | 0.080 |
| | 0.20 | 0.943±0.002 | 0.935±0.002 | 0.949±0.001 | 0.942±0.001 | 0.940±0.001 | **0.953±0.001** | 0.4 | 0.950±0.001 | 0.040 |
| | 0.30 | 0.939±0.001 | 0.929±0.001 | 0.948±0.001 | 0.939±0.001 | 0.938±0.001 | **0.953±0.001** | 0.5 | 0.952±0.001 | 0.040 |
| | 0.40 | 0.932±0.002 | 0.912±0.027 | 0.946±0.001 | 0.937±0.001 | 0.936±0.001 | **0.950±0.001** | 0.5 | 0.949±0.001 | 0.040 |
| CIFAR-10 | 0.10 | 0.934±0.002 | 0.931±0.002 | 0.955±0.002 | 0.946±0.002 | 0.945±0.002 | **0.956±0.001** | 0.7 | **0.956±0.002** | 0.060 |
| | 0.20 | 0.908±0.002 | 0.912±0.002 | 0.952±0.003 | 0.941±0.002 | 0.940±0.002 | 0.952±0.002 | 0.7 | **0.953±0.002** | 0.080 |
| | 0.30 | 0.862±0.003 | 0.879±0.002 | 0.944±0.002 | 0.934±0.002 | 0.933±0.002 | 0.945±0.002 | 0.7 | **0.945±0.002** | 0.080 |
| | 0.40 | 0.802±0.003 | 0.834±0.003 | 0.931±0.002 | 0.925±0.002 | 0.923±0.002 | 0.933±0.002 | 0.8 | **0.934±0.002** | 0.100 |
| CIFAR-100 | 0.10 | 0.833±0.004 | 0.829±0.004 | 0.853±0.004 | 0.827±0.005 | 0.825±0.005 | 0.855±0.004 | 0.5 | **0.856±0.004** | 0.020 |
| | 0.20 | 0.815±0.003 | 0.815±0.003 | 0.850±0.004 | 0.815±0.005 | 0.812±0.005 | 0.852±0.004 | 0.6 | **0.854±0.003** | 0.020 |
| | 0.30 | 0.785±0.004 | 0.789±0.005 | 0.844±0.005 | 0.800±0.005 | 0.797±0.006 | 0.844±0.004 | 0.7 | **0.846±0.005** | 0.020 |
| | 0.40 | 0.746±0.004 | 0.755±0.003 | 0.833±0.004 | 0.779±0.005 | 0.774±0.005 | 0.834±0.003 | 0.7 | **0.834±0.004** | 0.020 |

In the main experiment of Tables 1 and 2, the corrupted datasets include pre-existing, unaccounted for label errors. While the usage of several datasets demonstrated the broad applicability and efficacy of the proposed loss functions, an additional experiment on a dataset without pre-existing label errors would enable a more precise measure of label error detection performance. To achieve this, a version of CIFAR-100 with test samples having human-verified label errors removed (Northcutt et al., 2021b) is used. The cleaned test set is artificially corrupted in the same manner as in the main experiment, and the Confident Learning error detection framework is once again applied; however, rather than using a k-fold approach over the entire dataset, model training is performed strictly on the non-cleaned training set, and evaluation is performed on the cleaned (and artificially corrupted) test set. Results averaged over three seeds are shown in Table 3 and form a direct comparison to the CIFAR-100 experiments shown in Table 1. The performance of all loss functions is improved, especially at lower corruption rates (improvement of nearly 0.1 in F1 score at $\eta = 0.1$). This improvement at $\eta = 0.1$ matches expectations given that the proportion of pre-existing label errors is more significant at lower artificial corruption rates. The most dramatic improvement is seen for PZ loss for all $\eta$, which performs best in all cases.

Table 3: Label error detections on the CIFAR-100 cleaned test set, averaged over three trials, measured using the F1 score ($\pm\sigma$). Performance is improved relative to Table 1, especially at lower $\eta$ and for PZ loss.

| $\eta$ | CE | FL | BL | PZ |
|---|---|---|---|---|
| 0.1 | 0.576±0.027 | **0.587±0.030** | 0.580±0.024 ($\gamma = 0.3$) | **0.590±0.020** ($c = 0.020$) |
| 0.2 | 0.670±0.013 | 0.671±0.016 | **0.693±0.013** ($\gamma = 0.3$) | **0.717±0.017** ($c = 0.015$) |
| 0.3 | 0.697±0.011 | 0.709±0.006 | **0.775±0.011** ($\gamma = 0.5$) | **0.780±0.013** ($c = 0.020$) |
| 0.4 | 0.728±0.005 | 0.729±0.002 | **0.824±0.003** ($\gamma = 0.7$) | **0.830±0.007** ($c = 0.025$) |

## 6.2 Ablation Studies

**Training Dynamics** To verify that the proposed loss functions affect gradients during training as they are designed to do, the gradient of loss, $\partial\mathcal{L}/\partial p_{\tilde{y}}$, is monitored during training for corrupted and clean samples. Box and whisker plots in Figure 3 indicate the distribution of gradients for corrupt (red) and clean (green) data at each epoch during training. A detailed (reduced vertical axis) view is shown in the top row, while the full view is shown in the bottom row. CIFAR-100 with $\eta = 0.4$ is used for this study. With Cross Entropy loss, the gradients of corrupt data are large and negative, which provides a strong signal to the optimizer to fit to these corrupt data (undesired, not robust), whereas with Blurry Loss, gradients of corrupt data are large and *positive*, steering away from these corrupt data, and with Piecewise-zero Loss, gradients of corrupt data are nearly all at zero, imparting minimal impact on training. Therefore, the proposed Blurry and Piecewise-zero Losses operate as intended by minimizing fitting to corrupt data.

Building on Figure 3, additional box and whisker plots in Figure 4 indicate the distribution of predicted probabilities during training. For the trained model to be useful for the downstream detection of samples likely to have label errors, the distributions of predicted probabilities for clean *vs.* corrupt data should be distinct; *i.e.*, we desire that clean data be assigned high predicted probabilities whereas corrupt data be assigned low predicted probabilities. With all loss functions, the predicted probabilities of corrupt data are less than those of clean data; however, the difference is far greater with the proposed losses, thereby explaining *how* the downstream label error detection frameworks benefit from the proposed loss functions.

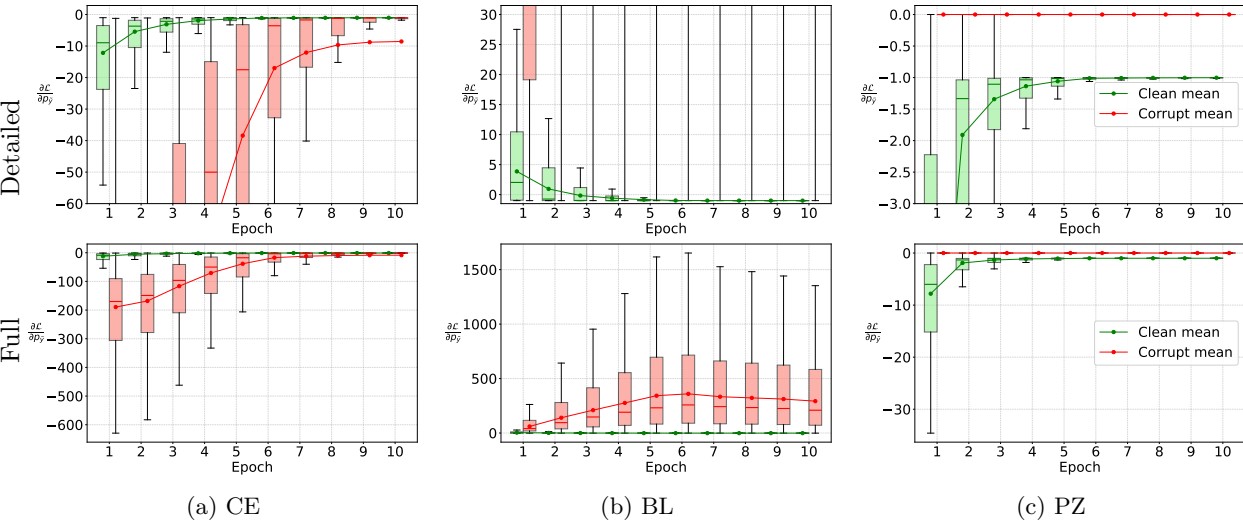

(a) CE          (b) BL          (c) PZ

Figure 3: Gradient distributions per epoch, confirm the proposed BL and FZ loss operate as intended and minimize model fitting to corrupt data. Results are shown for CIFAR-100 at $\eta = 0.4$, with a detailed view in the top row and full vertical axes in the bottom row. Plots for corrupt data are shown in red and for clean data in green. Trimmed means (15th/85th percentiles) are used to avoid the influence of outliers, and 1.5 IQR whiskers are used. With CE, the gradients of corrupt data are large and negative, which provides a strong signal to the optimizer to fit to these corrupt data, whereas with BL, gradients of corrupt data are large and *positive*, steering away from these corrupt data, and with PZ, gradients of corrupt data are nearly all at zero, imparting minimal impact on training.

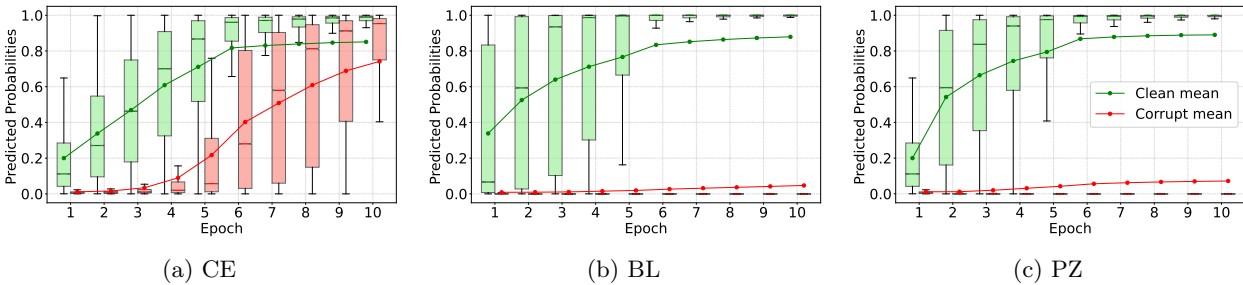

Figure 4: Predicted probability distributions per epoch, confirm the proposed BL and FZ loss operate as intended and better separate clean and corrupt samples. Results are shown for CIFAR-100 at $\eta = 0.4$. 1.5 IQR whiskers are used. With all loss functions, the predicted probabilities of corrupt data (red) are less than those of clean data (green); however, these differences are far greater with BL and PZ loss, leading to improved downstream detection of label errors.

**Non-uniform Corruption**  The main experiment included only *uniform* artificial corruption. To better understand the performance of the proposed loss functions under the presence of *non-uniform* label errors, a study involving the CIFAR-10N and CIFAR-100N (Wei et al., 2022) datasets with real-world human annotation errors is performed. This study follows the same format as the main experiment. Results averaged over five random seeds are shown in Table 4 against baseline loss functions, and for three variations of the CIFAR-N datasets, as described in Section 5. Rows are ordered with increasing corruption rate. Notice that the proposed loss functions are best (and second best) performing in all cases, and the optimal parameter setting tends to increase with corruption rate, congruent with the results of ablation Figure 8 in Section A.2.

Table 4: Label error detections on the CIFAR-10N and CIFAR-100N datasets with non-uniform corruption, averaged over five random trials, measured using the F1 score ($\pm\sigma$). The proposed BL and PZ loss functions continue to exceed baseline performance, with non-uniform corruption.

| Dataset | CE | Focal | GCE | ANL-CE | ANL-FL | BL | $\gamma$ | PZ | $c$ |
|---|---|---|---|---|---|---|---|---|---|
| CIFAR-10N, $\eta \approx 0.1$ | $0.662 \pm 0.004$ | $0.634 \pm 0.002$ | $0.698 \pm 0.002$ | $0.652 \pm 0.006$ | $0.647 \pm 0.003$ | $\mathbf{0.698 \pm 0.004}$ | 0.7 | $\underline{\mathbf{0.704 \pm 0.006}}$ | 0.08 |
| CIFAR-100N, $\eta \approx 0.4$ | $0.700 \pm 0.002$ | $0.684 \pm 0.002$ | $0.741 \pm 0.002$ | $0.721 \pm 0.002$ | $0.721 \pm 0.003$ | $\mathbf{0.742 \pm 0.003}$ | 0.8 | $\underline{\mathbf{0.745 \pm 0.003}}$ | 0.10 |
| CIFAR-10N, $\eta \approx 0.4$ | $0.768 \pm 0.001$ | $0.769 \pm 0.002$ | $0.837 \pm 0.002$ | $0.865 \pm 0.001$ | $0.865 \pm 0.002$ | $\underline{\mathbf{0.867 \pm 0.001}}$ | 0.9 | $\mathbf{0.866 \pm 0.001}$ | 0.15 |

**Large-scale Real-world Datasets**  To evaluate whether the proposed robust losses continue to provide benefits beyond small-to-medium benchmark settings, we additionally replicated the main experimental pipeline on two large-scale real-world noisy datasets: Food-101N ($\eta \approx 0.2$) (Lee et al., 2018) and Clothing-1M ($\eta \approx 0.4$) (Xiao et al., 2015). As shown in Table 5, for both datasets the proposed loss functions outperform the baselines when comparing either F1 or Balanced Accuracy scores. Similar to previous experiments, the optimal parameter setting tends to increase with corruption rate, offering insight to parameter selection for other datasets. Graphing the results for a wide range of parameters for experiments on the Food101-N dataset in Figure 5 shows that performance is improved for any setting across a broad range of parameter values rather than requiring narrow tuning. Full parameter sweep results for Food101-N and Clothing-1M are provided in Section A.2, Figures 9 and 10, which further confirm a wide range of parameters yield improved performance.

**Alternative Label Error Detection Framework**  The main experiment used Confident Learning as the label error detection framework. To demonstrate the effectiveness and versatility of the proposed loss functions when used with other frameworks, an ablation study using the AUM (Pleiss et al., 2020) framework is performed on the CIFAR-100 dataset. Resulting label error detection F1 scores averaged over three random seeds are summarized in Table 6 and are directly comparable to the CIFAR-100 results of Table 1. The label error detection performance exceeds that of Confident Learning, and the proposed loss functions improve upon the baseline.

Table 5: Label error detection on large-scale datasets with real world label noise, Food-101N and Clothing-1M. Detection results are averaged over 5 seeds, measured by F1 score and Balanced Accuracy ($\pm\sigma$). The proposed loss functions continue to outperform baseline losses for large datasets.

| Dataset | $\eta$ | Metric | CE | FL | GCE | ANL-CE | ANL-FL | BL | $\gamma$ | PZ | $c$ |
|---|---|---|---|---|---|---|---|---|---|---|---|
| Food-101N | $\approx 0.2$ | F1 | 0.436±0.001 | 0.415±0.004 | 0.458±0.007 | 0.344±0.003 | 0.348±0.002 | **0.498±0.004** | 0.5 | **0.524±0.004** | 0.07 |
| | | BA | 0.654±0.001 | 0.640±0.002 | 0.711±0.006 | 0.569±0.005 | 0.575±0.003 | **0.724±0.009** | 0.6 | **0.753±0.004** | 0.09 |
| Clothing-1M | $\approx 0.4$ | F1 | 0.530±0.004 | 0.484±0.004 | 0.655±0.001 | 0.518±0.032 | 0.520±0.044 | **0.687±0.003** | 0.9 | **0.695±0.004** | 0.3 |
| | | BA | 0.670±0.002 | 0.650±0.002 | 0.734±0.001 | 0.529±0.031 | 0.535±0.043 | **0.747±0.002** | 0.9 | **0.753±0.004** | 0.2 |

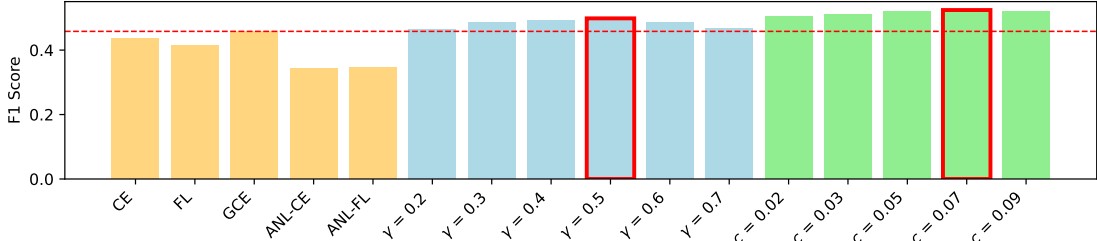

Figure 5: Label error detection F1 scores on Food-101N ($\eta \approx 0.2$). Columns in orange represent the baseline loss functions, with the dashed red line showing the top baseline performance. BL is shown in blue, and PZ is green. For all tested parameter values, BL and PZ outperform traditional loss functions (Cross Entropy and Focal Loss), and outperform SOTA robust loss functions over a wide range of parameter values.

**Training Dynamics through AUM Statistic** As was explained in Section 3, the AUM statistic (Pleiss et al., 2020) measures the difference between the logit values for a training example's as-labelled class and its highest other class, averaged over the course of training. Correctly labelled samples tend to exhibit positive AUM values, while mislabelled samples tend to have significantly lower AUM values due to conflicting gradient updates for samples of the same class. To better understand how the proposed loss functions affect the training dynamics on corrupt *vs.* clean data, the AUM is measured for all samples, and the AUM distributions for corrupt and clean data are compared. Figure 6 illustrates the AUM distributions for training with CE, BL, and PZ loss on the CIFAR-100 dataset with artificial corruption at $\eta = 0.1$ (following the same method used throughout this paper). The mean of each distribution is indicated using a dashed vertical line. To quantitatively understand the impacts of training using the proposed loss functions, two measures of dissimilarity in distributions are computed: Cohen's d (Cohen, 2013) (Effect size) and Wasserstein distance (Kantorovich, 1960; Vaserstein, 1969), both of which are sensitive to differences in distribution position and width. Note that an ideal robust training scheme would result in a large difference in the AUM distributions of corrupt and clean samples.

Under CE training (Figure 6a), clean and corrupted samples exhibit distinct but moderately overlapping distributions. Although separation exists, the overlap reflects CE's tendency to fit both clean and noisy labels, leading to partial memorization of mislabelled samples.

Training with Blurry Loss (Figure 6b), produced a markedly improved separation. Quantitatively, a Cohen's d of 3.43 (*vs.* 3.20 for CE), and Wasserstein distance of 8.40 (*vs.* 6.11 for CE) were measured, indicating improved separation of corrupt and clean data. This increased separation of the distributions supports our intention and hypothesis that BL de-weights hard-to-classify (likely mislabelled) samples, reducing the extent to which the model fits to label errors. Consequently, the discrimination between clean and noisy samples becomes more pronounced, enhancing the effectiveness of AUM-based label error detection.

The most dramatic effect was observed when using Piecewise-zero Loss (Figure 6c). Here, Cohen's d increased to 3.82 (*vs.* 3.20 for CE), and the Wasserstein distance to 13.72 (*vs.* 6.11 for CE), indicating greater separation than both CE and BL. The sharp cutoff mechanism of PZ effectively prevents low-confidence samples from contributing to weight updates during training, almost eliminating their influence. As a result, mislabelled examples tend consistently to have highly negative AUM values, leading to the clearest distributional

Table 6: Label error detections using the AUM framework and with the CIFAR-100 dataset, averaged over three random trials, measured using the F1 score ($\pm\sigma$). The proposed losses continue to perform well using AUM, in addition to with CL (Tables 1 and 2).

| Loss Function | $\eta = 0.1$ | $\eta = 0.2$ | $\eta = 0.3$ | $\eta = 0.4$ |
|---|---|---|---|---|
| CE | $0.7452 \pm 0.0071$ | $0.8638 \pm 0.0042$ | $\mathbf{0.9019 \pm 0.0021}$ | $\mathbf{0.9197 \pm 0.0024}$ |
| BL ($\gamma = 0.05$) | $\mathbf{0.7591 \pm 0.0180}$ | $\mathbf{0.8692 \pm 0.0032}$ | $\underline{0.9076 \pm 0.0031}$ | $\underline{\mathbf{0.9261 \pm 0.0005}}$ |
| PZ ($c = 0.025$) | $\underline{\mathbf{0.8817 \pm 0.0074}}$ | $\underline{\mathbf{0.8903 \pm 0.0085}}$ | $0.8914 \pm 0.0044$ | $0.8856 \pm 0.0125$ |

(a) CE        (b) BL        (c) PZ

Figure 6: AUM distributions for artificially corrupted and clean samples in CIFAR-100 with corruption at $\eta = 0.1$ for training with CE (a), BL (b), and PZ loss (c). Greater separation in distributions indicates greater robustness of the loss function to mislabelled samples during training. Separation is measured using Cohen's d {3.20 (CE) < 3.43 (BL) < 3.82 (PZ)} and Wasserstein distance {6.11 (CE) < 8.40 (BL) < 13.72 (PZ)}.

separation among all loss functions tested. Indeed, the F1 score notably increased by more than 13% relative to CE, as was shown in Table 6.

Both BL and PZ loss functions substantially improve the margin-based separability of clean versus mislabelled samples compared to CE. These results validate our design objectives: the proposed loss functions mitigate the harmful fitting to noisy labels by selective de-weighting or exclusion. This enhanced separability directly translates into improved performance for label error detection frameworks like AUM and Confident Learning.

**Further Studies** Additional investigations and ablation studies are provided in Section A.2, including a study observing the precision-recall trade-off and their impact on F1 score, in Figure 7, a study investigating the impacts of the proposed loss function parameters across various datasets, in Figures 8 to 10, and an ablation study on the effects of the delay parameter, $d$, in Figure 11.

# 7 Conclusion and Limitations

This work introduces and provides an in-depth empirical evaluation of two novel loss functions, Blurry Loss and Piecewise-zero Loss, for improving model robustness and the downstream detection of label errors. By de-weighting or ignoring samples likely to be mislabelled, these losses enhance the performance of label error detection frameworks across a variety of artificial corruption and real-world label-noise settings. Extensive experiments and ablation studies demonstrate their broad applicability and competitive performance relative to existing robust loss functions. One limitation of the experiments is that parameter sweeps were not performed for the existing robust loss functions; however, their recommended settings for each dataset were used wherever possible. Similarly, there is no universally optimal parameter setting for the proposed loss functions. These findings support the use of such loss functions as practical tools for improving data quality and model reliability in supervised learning scenarios fraught with label errors.

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

# A    Technical Appendices and Supplementary Material

## A.1    Additional Experimental Details

Further information building upon the experimental details of Section 5 is given here.

**Models and Optimizers**    For MNIST and Fashion-MNIST, we adopt a minimal convolutional neural network consisting of two convolutional layers followed by two fully connected layers interleaved with dropout, totalling approximately 1.2 million trainable parameters. Under a ten-epoch training schedule, this architecture reliably exceeds 99% test accuracy (near state-of-the-art) (Kadam et al., 2020).

For CIFAR-10, CIFAR-100, and their noisy variants (CIFAR-10N/CIFAR-100N), we fine-tune an ImageNet-pretrained ResNet-34 (He et al., 2016) modified only by replacing its initial $7 \times 7$, stride-2 convolution with a $3 \times 3$, stride-1 convolution and removing the following max-pool layer, preserving spatial resolution on $32 \times 32$ inputs. We train for ten epochs under identical conditions using the Adam optimizer (Kingma, 2014) with an initial learning rate of $1 \times 10^{-4}$; a scheduler reduces the learning rate by a factor of 0.1 at epoch 5 for CIFAR-10 (factor 0.2 for CIFAR-100). Under this setup, the model attains 93.5% test accuracy on CIFAR-10 and 74.5% on CIFAR-100.

All experiments utilize the Adam optimizer, leveraging its adaptive learning capabilities without requiring additional tuning.

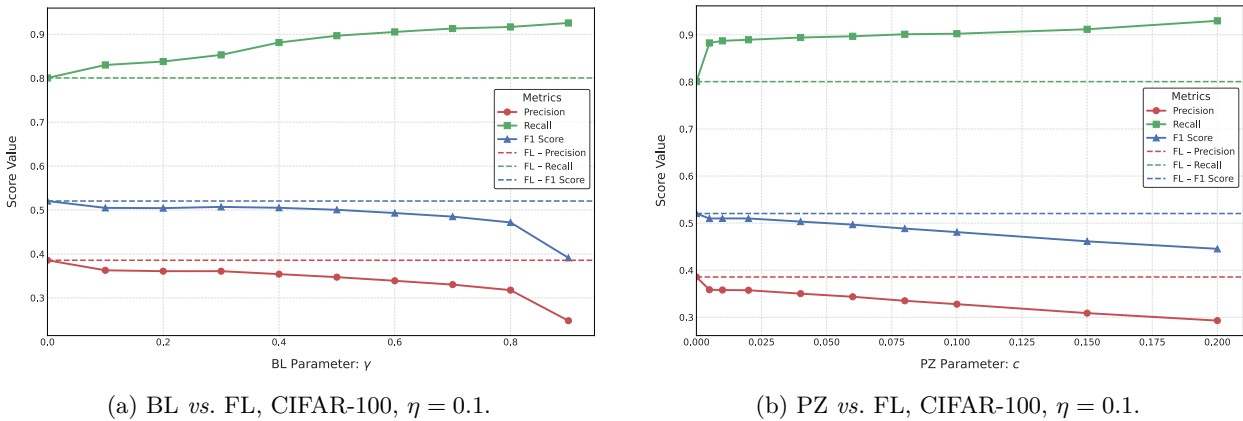

(a) BL *vs.* FL, CIFAR-100, $\eta = 0.1$.
(b) PZ *vs.* FL, CIFAR-100, $\eta = 0.1$.

Figure 7: FL (dotted lines) results in better F1 scores than BL (a) or PZ (b) (solid lines) at $\eta = 0.1$ on the CIFAR-100 dataset. This is a result of conservative detection, whereby fewer samples are detected and precision is maintained. Through tuning the parameters of the proposed loss functions, performance can be made fairly similar to that of FL, or recall can be increased dramatically while only marginally reducing precision, which may be considered worthwhile at low corruption rates.

**Compute Resource Requirements**  All experiments were conducted on a compute cluster using an NVIDIA Tesla V100-SXM2-16GB GPU. The main Confident Learning experiments required 6 GB of memory, and each loss function and parameter combination required on average 10 minutes (5 minutes for MNIST and Fashion-MNIST, 15 minutes for CIFAR-10 and CIFAR-100). Experiments on the cleaned CIFAR-100 test set followed the same process, requiring only 3 GB memory. When additionally storing the per-epoch gradient information, for MNIST 3 GB of memory and 1 hour was required, and 6 GB of memory and 4 hours for CIFAR-100. The CIFAR-N experiments required 15 minutes and 6 GB of memory. The AUM experiments required 7 GB of memory as well as 2.5 hours for each loss function and parameter test conducted. The experiments on Food-101N each took 8 hours to complete, and 13 hours for Clothing-1M experiments. The complete set of experiments required approximately 3400 hours of compute, with an additional $\sim 400$ hours for preliminary experimentation.

The Confident Learning experiments were conducted using the Python library `cleanlab` v2.5.0, from `https://github.com/cleanlab/cleanlab`, as introduced in Northcutt et al. (2021a) under an AGPL-3.0 license. The AUM experiments were conducted using the python library `aum` v1.0.2, from `https://github.com/asappresearch/aum` as introduced in Pleiss et al. (2020) under an MIT license.

## A.2   Additional Experiments and Studies

**Precision-Recall Trade-off in F1 Score**  Precision and recall (the two harmonic components of the F1 score) are explored for BL and PZ, and compared to FL in Figure 7, for the CIFAR-100 dataset at $\eta = 0.1$, a case selected since the F1 score of FL exceeds that of BL and PZ. Observe that by tuning the parameters of the proposed loss functions, recall is increased significantly, but at the cost of some precision. Because F1 is a harmonic mean, the small decrease in precision dominates the larger increase in recall, and the score is decreased. Again, this does not necessarily indicate less desirable behaviour at lower corruption rates, since having improved recall may be worthwhile at the expense of having relatively few false-positives. To capture a more reasonable objective at lower corruption rates, Balanced Accuracy (mean of sensitivity and specificity) is reported in Table 2, in which the proposed BL and PZ perform strongly throughout.

**Loss Function Parameters**  To better understand the parameters of the proposed loss functions, an ablation study on these parameters is performed using a similar experimental setup as the main experiment. For each dataset and artificial corruption rate, the optimal parameter setting is identified. For these experiments, the delay, $d$, is set to 0 for BL and 1 for PZ loss. Results for MNIST and CIFAR-100 datasets are shown in Figure 8 as a heatmap of F1 score, along with a comparison to the baseline loss functions.

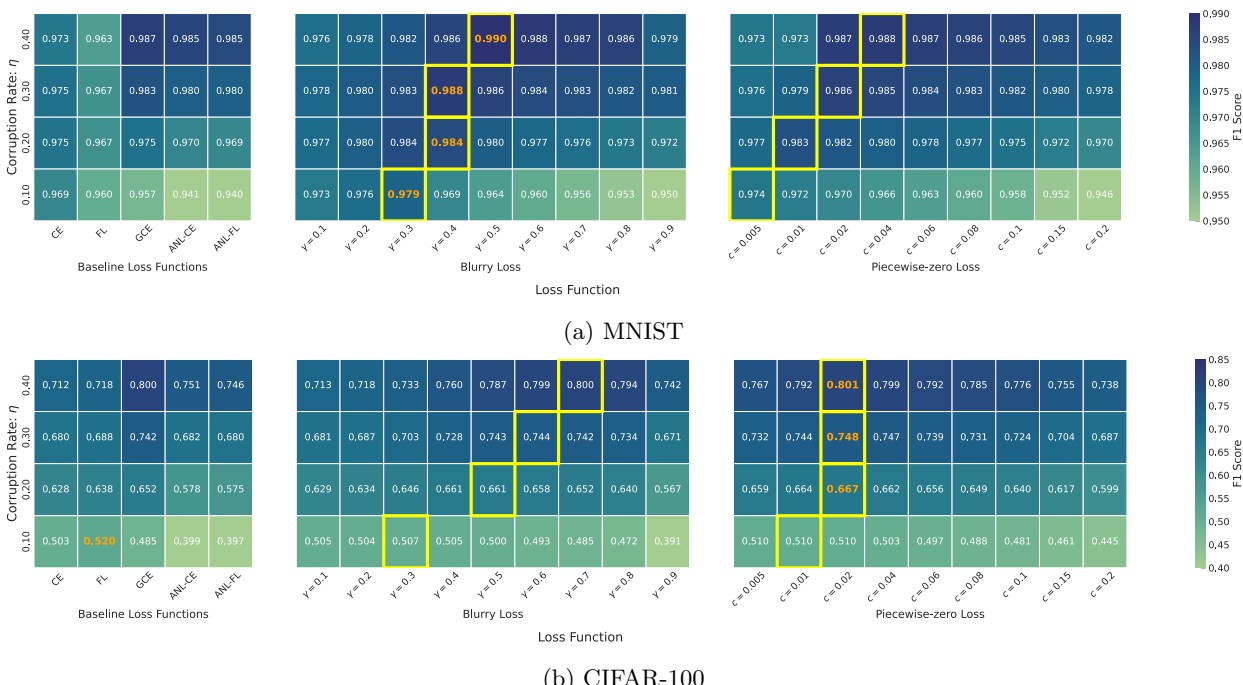

(a) MNIST

(b) CIFAR-100

Figure 8: Heatmap showing variation in F1 score as loss function parameters are varied, for MNIST (a) and CIFAR-100 (b). In the left blocks, results for baseline loss functions are shown. Two separate blocks are shown for the proposed loss functions, plotted over a range in their parameters. Best results are indicated with bolded orange text. To illustrate the trend in optimal parameter settings for the proposed loss functions, yellow boxes mark the best performing parameter setting for each row. Note that delay, $d$, is set to 0 for BL and 1 for PZ loss.

For each corruption rate (row), the best performing configuration is indicated with bolded orange text. To illustrate the trend in optimal parameter settings for the proposed loss functions, yellow boxes mark the best performing settings. Observe that optimal settings of both $\gamma$ and cutoff, $c$, increase with increased corruption rate. This trend is consistent across datasets; however, more complicated / difficult datasets, such as CIFAR-100 relative to MNIST, seem also to demand greater settings of $\gamma$ (but not $c$). Note that larger settings for $\gamma$ and $c$ correspond to the proposed loss functions 'ignoring' a greater range of predicted probabilities.

We additionally repeat this parameter ablation procedure on the two large-scale real-world noisy-label datasets, Food-101N and Clothing-1M, to assess whether the observed trends persist beyond artificially corrupted benchmarks. Since corruption in these datasets arises naturally and no precise corruption rate $\eta$ is provided, we instead perform a grid search over $\gamma$ and cutoff $c$ and select the best-performing configuration based on F1 and Ballanced Accuracy scores. Results are shown in Figure 9 (Food-101N; with F1 results repeated from Figure 5 of Section 6.2) and Figure 10 (Clothing-1M), where the parameter landscapes again indicate that larger values of $\gamma$ and $c$ (corresponding to ignoring a larger range of low-confidence predictions) tend to yield improved robustness in the presence of heavy label noise. For both large-scale datasets, a comparatively broad set of near-optimal parameter configurations exist, suggesting that the proposed losses are not overly sensitive to precise parameter tuning.

**Delay** Delay, $d$, is introduced to allow the model to generalize prior to training with the proposed loss functions. This is especially important for PZ loss, which, at the initial stages of training, outright 'ignores' a large proportion of the dataset. To better understand how best to set the delay, an ablation study on this parameter is performed. Again, label errors are detected and performance is measured with F1 score, while $d$

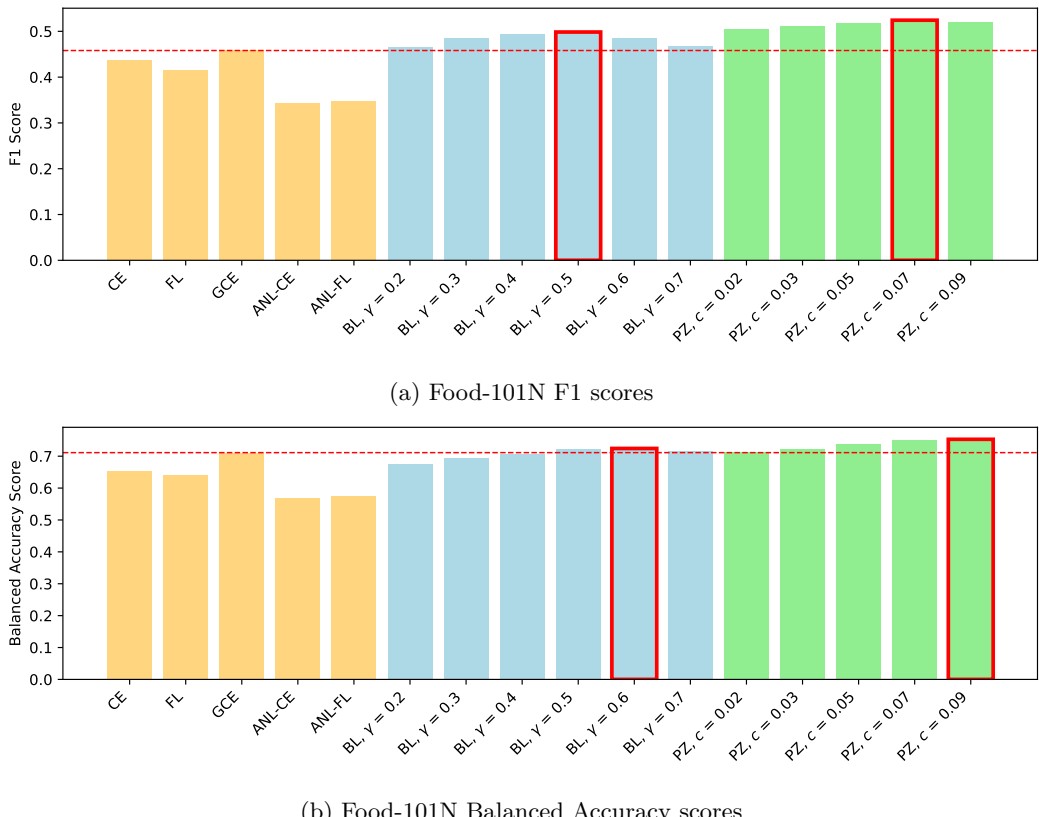

(a) Food-101N F1 scores

(b) Food-101N Balanced Accuracy scores

Figure 9: Label error detection results on Food-101N ($\eta \approx 0.2$). F1 results are repeated from Figure 5 shown previously in Section 6.2. Dashed red line shows the top baseline performance. For all tested parameter values, BL and PZ outperform traditional loss functions (Cross Entropy and Focal Loss), and have a wide range of values which outperform SOTA robust loss functions.

is varied. This study was performed across all the main experiment datasets and corruption rates, with fairly consistent results.

Results for CIFAR-100, the most complex dataset which we artificially corrupt, are given in Figure 11, for corruption rates ranging from $\eta = 0.1$ to $0.4$. Trends for various cutoff, $c$, settings are plotted. Observe that there is a large jump in F1 from $d = 0$ to $d = 1$ (for all $\eta$), indicating that some delay is highly beneficial and allowing the model to somewhat generalize early in training is necessary for good application of PZ loss. Additionally, the best performance is achieved for $d = 1$ (for an appropriate setting of $c$), and F1 scores tend to towards the CE baseline (poorer performance) as delay is further increased. This indicates that having a greater number of epochs trained with PZ loss improves label error detection once the model is sufficiently generalized (one epoch with CE), and perhaps even that detrimental fitting to the erroneous data occurs with further training with CE. Furthermore, notice that the degree of downwards trending relates to the corruption rate, indicating that high corruption rates necessitate a greater need for training with PZ loss to achieve high label error detection performance.

Although not shown here, a similar preliminary study was performed for BL, which indicated that having no delay was always best, and thus delay is suggested only for PZ loss. This is likely a result of PZ loss being far more harsh than BL (outright ignoring of data rather than simply having positive gradients for samples with low predicted probabilities).

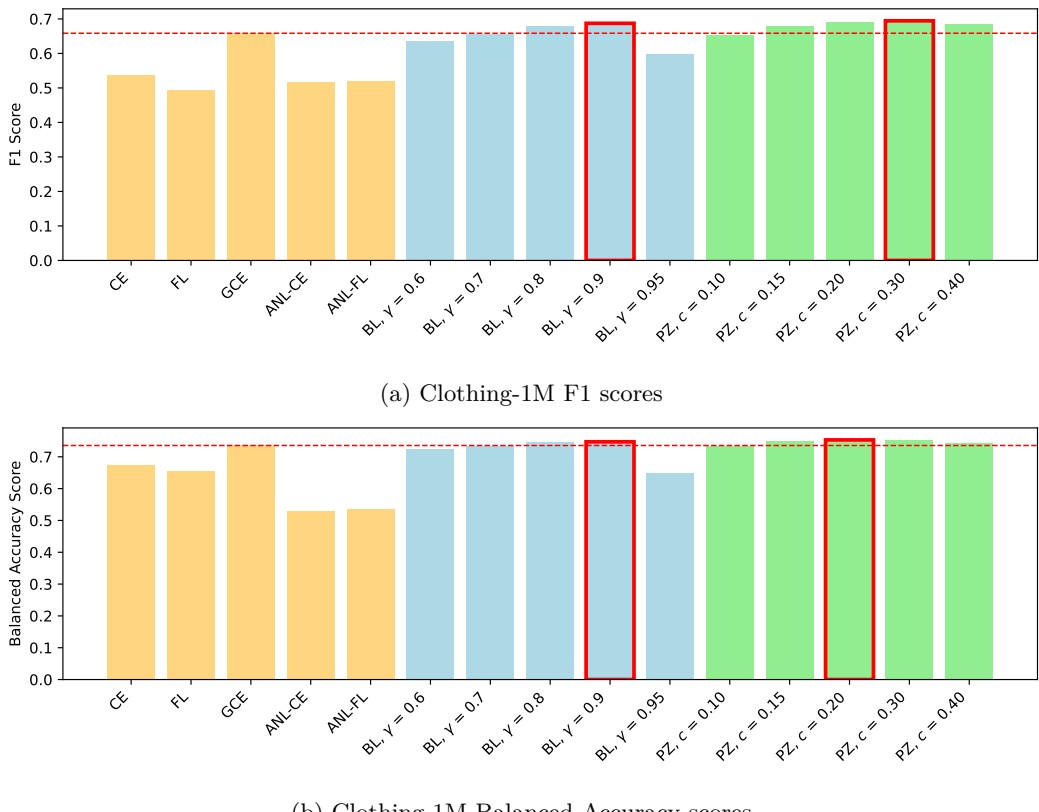

(a) Clothing-1M F1 scores

(b) Clothing-1M Balanced Accuracy scores

Figure 10: Label error detection results on Clothing-1M ($\eta \approx 0.4$). Dashed red line shows the top baseline performance. For all tested parameter values, BL and PZ outperform traditional loss functions (Cross Entropy and Focal Loss), and have a wide range of values which outperform SOTA robust loss functions.

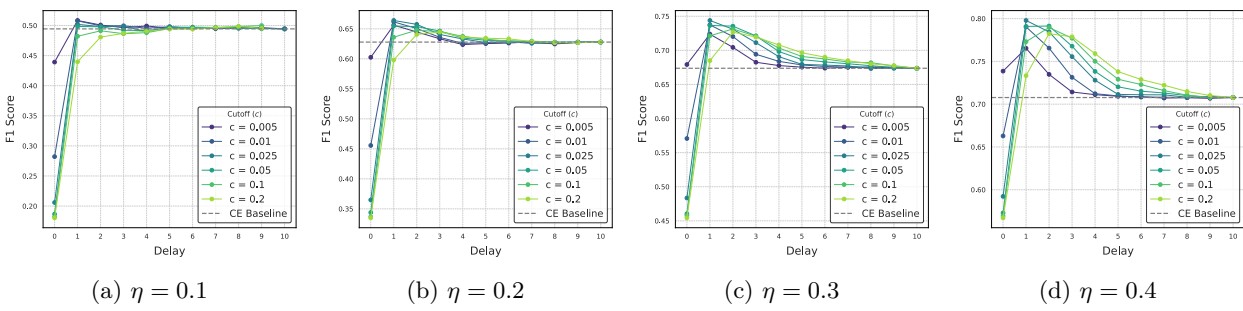

(a) $\eta = 0.1$       (b) $\eta = 0.2$       (c) $\eta = 0.3$       (d) $\eta = 0.4$

Figure 11: Detection F1 score *vs.* Delay, $d$, for PZ loss on CIFAR-100 at $\eta$ ranging from 0.1 to 0.4 in panels (a) to (d). Observe that the best performance comes at $d = 1$ in all cases, for an appropriate setting of the cutoff, $c$.

