# OpenReview forum: "Hard Samples, Bad Labels: Robust Loss Functions That Know When to Back Off"
_TMLR — Rejected by TMLR_

### Review · Reviewer_r5kZ · 2026-03-13

**Summary Of Contributions:**

**Overview:** This paper focuses on the important problem of label error (or label noise) detection. Based on the idea of mislabelled examples should contribute less instead of more, the authors propose two novel loss functions, *Blurry Loss* (BL) and *Piecewise-zero Loss* (PZ), each of which has well-designed formulation and promising statistical properties. Experimental results on label error detection task show the superiority of BL and PZ, which is further strengthened by diverse ablations.

**Strengths:**

(1) The method shows novelty. I recognize the author's creativity on designing the loss formulation. It arises from the idea of deweighting the mislabelled examples. Take BL as an example, BL is a convex function and has one positive maximum point. In the initial stage, if the loss falls to the left side, it allow the model to go into an optimization that decreases them, which differs significantly with standard CE optimization. This makes BL more robust, especially with high noise ratio.

(2) The paper is reader-friendly. The authors laid a lot of background information (Tasks, Standard loss functions, Toy examples, etc)  before introducing the main text. They illustrate the statistical property of BL and PZ with multiple diagrams, clear formulations and detailed explanations.

(3) The experimental results are promising. I should also mention that they are incredibly detailed when explaining the experimental settings, which surely makes reader relieved and confident about their results. Plus, I did try the provided anonymous codebase and Jupyter notebook guide. I acknowledge the paper's reproductivity.

**Weaknesses:**

(1) I beg to argue that no significant contributions are made. The paper does not consider the label-noise learning (noisy label learning: NLL) study while they definitely should've. Label error detection is an important task, however, it kinda strikes me as an upstream task. I am very curious what BL/PL will bring in downstream NLL tasks, where the field is broader and the learning algorithms are much more competitive. For example, there's many sample selection/weighting methods in NLL [1][2], which is very similar compared with label error detection but beyong. Even the idea of PZ, filtering the influence of mislabelled examples, is not uncommon [3].

(2) I suggest the authors re-investigate the field of label error detection (sample selection). In fact, there are more advanced methods beyond loss design. For example, [4] leverages VLM to detect and correct noisy labels. Maybe some comparisons would make the paper more valuable.

(3) The paper mainly focuses on instance-independent noise, which I think is promising, but not practical enough. I suggest the authors expand the setting to instance-dependent noise, which I reckon as a more practical and meaningful setting. See [5] for details.

**Reference**

[1] Shu J, Yuan X, Meng D, et al. Cmw-net: Learning a class-aware sample weighting mapping for robust deep learning[J]. IEEE Transactions on Pattern Analysis and Machine Intelligence, 2023, 45(10): 11521-11539.

[2] Li Y, Han H, Shan S, et al. Disc: Learning from noisy labels via dynamic instance-specific selection and correction[C]//Proceedings of the IEEE/CVF conference on computer vision and pattern recognition. 2023: 24070-24079.

[3] Wei Q, Sun H, Lu X, et al. Self-filtering: A noise-aware sample selection for label noise with confidence penalization[C]//European Conference on Computer Vision. Cham: Springer Nature Switzerland, 2022: 516-532.

[4] Wei T, Li H T, Li C S, et al. Vision-language models are strong noisy label detectors[J]. Advances in Neural Information Processing Systems, 2024, 37: 58154-58173.

[5] Xia X, Liu T, Han B, et al. Part-dependent label noise: Towards instance-dependent label noise[J]. Advances in neural information processing systems, 2020, 33: 7597-7610.

**Additional Comments:**

See above.

**Audience:**

Yes

**Audience Explanation:**

Yes. Label noise learning is a long-standing inevitable challenge in machine learning community. Their findings, especially the way they ilustrate their findings is good.

Also, BL and PZ is light-weight loss function module and might be seamlessly integrated into existing pipelines.

**Broader Impact Concerns:**

The proposed loss functions are based on the idea that mislabelled examples appear as outliers thus assigning them lower predicted probabilities. The method risks inadvertently pruning valid-but-rare data points. This could exacerbate model bias/fairness. The authors might add a brief discussion on how to might mitigate the risk of disproportionately filtering out rare but correct data when using these robust loss functions.

**Claims And Evidence:**

Yes

**Claims Explanation:**

Yes. The authors provide extensive evidence to support their claims, regarding the efficacy of BL and PZ. The formulations are clear defined and explained.

**Requested Changes:**

**Critical:**

(1) The paper lacks exploring the broader and highly competitive field of Noisy Label Learning (NLL). I suggest the authors should include a discussion comparing their proposed methods to existing sample selection and weighting algorithms in the NLL studies (e.g., Shu et al., 2023; Li et al., 2023; Wei et al., 2022, see more in summary section). Specifically, the idea of filtering the influence of mislabelled examples is well-explored in NLL (the comparison with PZ is important). This would give solid contributions.

**Others:**

(1) Comparison with advanced detection methods: Expanding the literature to include recent advancements, such as using VLMs for detecting and correcting noisy labels (e.g., Wei et al., 2024, see more in summary), would elevate the paper's value.

(2) Evaluation on Downstream Tasks: It would greatly strengthen the submission to include an experiment (or at least a robust discussion) showing the downstream performance of a model trained after the dataset has been pruned using BL/PZ versus standard baselines.

(2) Extending the setting to instance-dependent label noise and adding more experimental results on this one would be helpful.

---

> ### Author Response · Authors · 2026-04-10
> **Response to review**
>
> Thank you for your detailed review, and special thanks for taking the time to explore our codebase. We have responded individually to each of your raised questions and concerns below.
>
> **Re: Integration with noise label learning:**
>
> Thank you for pointing out similar concepts found in the field of NLL. Note that the goals in label error detection and NLL are subtly different: we seek to identify and propose samples likely to have label errors (even if some are correctly labelled, with trade-offs between precision and recall based on one’s own tolerance in their application) vs. LNL where one seeks a model capable of correctly classifying samples in spite of training on noisy data. To the best of our knowledge, these methods have not been applied to the task of label error detection. Nonetheless, we thank you for the suggestion to apply BL/PZ in NLL tasks in future work.
>
> **Re: Reviewing additional label error detection methods:**
>
> Thank you for the suggestion to broaden our Literature Review / Background to include some of these methods that you have listed. Note that our paper focuses on loss function design rather than comparisons between label error detection frameworks and approaches. Making an experimental comparison would be outside the scope of this paper; however, brief discussion of these types of approaches may indeed elevate the background of our paper.
>
> **Re: Instance-dependent noise:**
>
> We agree that instance-dependent noise is a critical and practical setting. While our artificial corruption experiments utilized instance-independent noise to establish a baseline, our evaluation on real-world datasets (e.g., CIFAR-N, Food-101N, and Clothing-1M) inherently addresses this concern. The errors in these datasets originate from human annotators and web scraping, making them naturally instance-dependent. Given our strong empirical results on these datasets, we are confident in the practical efficacy of our methods and will explicitly clarify this distinction in the revised manuscript.
>
> **Re: Comparison to NLL methods:**
>
> Thank you for this suggestion; we do see the similarity in these concepts. Although the scope of our paper is on loss function design and label error detection, relating the design of our loss functions to these existing concepts from NLL would be valuable. This discussion will be included in the revised manuscript.
>
> **Re: Evaluation on downstream tasks:**
>
> We appreciate your suggestion to evaluate downstream performance. While a comprehensive downstream evaluation falls outside the primary scope of this study, which centers on the detection of label errors, it is well-established in the literature that mitigating training set corruption directly improves model generalization and downstream efficacy (Northcutt et al., 2021b; Song et al., 2022). Furthermore, we emphasize that maximizing downstream performance requires more than simply pruning all flagged samples, as blind removal risks introducing bias (as you have indicated), hindering downstream learning. Rather, the optimal application of our robust loss functions is to first identify suspect samples for subsequent expert review and correction, thereby yielding a cleaned dataset. In extreme cases of high corruption, it may be more practical to broadly prune the data, thereby avoiding effectively re-annotating a large portion of the dataset. We would be happy to incorporate a robust discussion of these downstream implications in the revised manuscript.
>
> **Re: Discussion on pruning rare samples:**
>
> Thank you for raising this point regarding model fairness and the risk of pruning rare but correct data. While our loss functions do ‘ignore’ difficult-to-classify samples to avoid fitting the model to samples that are likely to be mislabelled, the intention behind our approach is for the downstream task of detecting label errors (even at the expense of proposing some rare but correctly-labelled data). Applying our loss functions for the purpose of NLL (learning on noisy data, without any task of pruning or correcting mislabelled samples) would very likely introduce model bias, and similarly, simply pruning a dataset without doing any re-annotation would also likely lead to biases.

---

### Review · Reviewer_LVxy · 2026-03-20

**Summary Of Contributions:**

This paper addresses the problem of supervised learning in the presence of incorrect labels. Label errors can severely degrade model performance, making this a critical challenge in neural network training. To this end, the authors propose two loss functions, the Blurry loss and the Piecewise-zero loss, designed to account for erroneous labels during training. Specifically, incorrect predictions are penalized less harshly, yielding smaller gradients in these regions and thereby preventing the model from forcefully fitting mislabeled samples. Experiments across several datasets with both simulated and realistic labeling errors demonstrate that the proposed methods achieve competitive performance compared to baselines.

**Audience:**

Yes

**Audience Explanation:**

The problem addressed by the authors is important, and the proposed loss functions are straightforward to apply in practice.

**Claims And Evidence:**

No

**Claims Explanation:**

While the paper has several strengths, the weaknesses outweigh them in my assessment. Below, I provide a detailed list leading to my decision:

**Strength**

- The paper is well written and the presentation (e.g. figures) is of high quality.
- The proposed loss functions are intuitive and easy to understand, making them easily applicable for practitioners.
- The experimental evaluation is extensive, covering a big number of datasets with both artificial and realistic label error scenarios.
- The results are averaged over 20 random trials, indicating strong statistical robustness.

**Weaknesses**

- **Insufficient motivation of the research gap.** The introduction does not properly explain why existing loss functions fall short in the presence of label noise. While related work is enumerated, there is no clear connection between it and the gap the authors claim to address. What specifically is problematic about current approaches, is it degraded performance, sensitivity to hyperparameters, or something else? Without this, it remains unclear how the proposed loss functions improve upon the state of the art.
- **Missing rationale for the design of the proposed loss functions.** The authors do not explain why *two loss functions* are proposed or justify *their specific design choices.* When should a practitioner choose the Blurry Loss over the Piecewise Zero Loss, and vice versa? A *clear recommendation or guideline distinguishing their respective strengths and trade-offs is expected* but absent. While empirical effectiveness is demonstrated, the lack of deeper insight into how and why these designs address the identified shortcomings is a notable limitation.
- **Fairness of the hyperparameter evaluation.** The authors perform a grid search for their approach but rely on recommended settings for the baselines. It is not stated which dataset the hyperparameter sensitivity analysis is conducted on. For a fair comparison, the grid search should be performed on a single dataset, and the resulting hyperparameters should be fixed across all remaining experiments. The chosen values should also be explicitly reported (e.g., "we use γ = 2 for all remaining experiments").
- **Missing comparison to related work.** The authors do not consider a significant body of related work. In particular, there is no discussion of the field of Learning with Noisy Labels (LNL), where approaches such as DivideMix [1] and ProMix [2] are well-established methods for handling label noise. A comparison against such methods would be expected. Furthermore, it would be insightful to investigate whether the proposed loss functions can be integrated into existing LNL pipelines and what improvements such integration could yield.
- **Unclear notion of robustness.** The authors argue that the gradient trends and predicted probability distributions in Figures 3 and 4 demonstrate improved robustness to mislabeled data. However, it is not explained what robustness means in this context or why these trends constitute evidence for it. A brief explanation of how the observed training dynamics connect to the claimed improvement would strengthen this argument.
- **Limited model architectures.** While using a ResNet is reasonable, the evaluation would be considerably strengthened by including experiments with a self-supervised pre-trained transformer architecture such as DINOv2, given the widespread adoption of transformers in current practice.
- **Misuse of the term "ablation study."** The section labeled as an ablation study also contains additional experimental results, including comparisons with baselines and extensions to realistic error datasets. Much of this content, particularly the realistic label noise experiments, should be part of the main evaluation.
- **Figure 1: Inaccurate terminology.** The term "ground truth label" is misleading, as the depicted label is incorrect. "Assigned label" would be more appropriate, since "ground truth" implies the label is correct.

[1] Li, Junnan, Richard Socher, and Steven C.H. Hoi. "DivideMix: Learning with Noisy Labels as Semi-supervised Learning." ICLR, 2020.

[2] Xiao, Ruixuan, Yiwen Dong, Haobo Wang, Lei Feng, Runze Wu, Gang Chen, and Junbo Zhao. "ProMix: Combating Label Noise via Maximizing Clean Sample Utility." IJCAI, 2023.

**Requested Changes:**

Given the weaknesses outlined above, I encourage the authors to address these in their response. I am open to revising my assessment if the raised concerns are adequately addressed.

---

> ### Author Response · Authors · 2026-04-10
> **Response to review**
>
> Thank you for your detailed review. We appreciate your acknowledgement of the statistical robustness and extensivity of our experiments. We have responded individually to each of your raised questions and concerns below.
>
> **Re: Motivation of research gap:**
>
> We would be happy to clarify the research gap. As is already well understood, conventional loss functions such as Cross Entropy and Focal Local loss are not robust to label errors and in fact can be quite sensitive to these. The existing SoTA robust loss functions (designed for learning with noisy labels, rather than the downstream detection of label error, for which our loss functions are designed) do improve upon baseline conventional loss functions; however, several shortcomings exist:
> 1. GCE is only partially robust to label errors and must compromise robustness with ability to converge during training, through parameter q, which controls the balance of GCE behaviour between CE (fast convergence but not robust) and MAE (fairly robust, but poor convergence).
> 2. ANL is more robust than GCE; however, it has four control parameters (difficult to apply to novel datasets) which balance the behaviour between the original ‘positive’ loss term and more robust variants (similar partial robustness as GCE).
>
> Our methods address these gaps by having far fewer parameters (in practice, just one each, since setting delay to 1 epoch for PZ is found to perform very well) and by being able to either ignore or push away from samples with label errors (as is clearly demonstrated in the results of Figures 3 and 4).
>
> **Re: Rationale for the design of the proposed loss functions:**
>
> The two loss functions were designed such that given a noisy label, the loss functions would de-weight the given label (BL), or ignore the sample altogether (PZ), in contrast to conventional loss functions, such as CE or FL, which would put more weight on these mislabeled samples. This is described near the end of the Introduction section, and visibly demonstrated and explained in Figure 1, where we show how different loss functions would work on a sample with a noisy label. A deeper analysis confirming our methods are functioning as hypothesized, and actively ignoring mislabeled samples is shown in 6.2 Ablation Studies – Training Dynamics. As both proposed loss functions consistently achieve better results than the compared baseline loss functions, we can recommend using either one in most situations and state that PZ achieved the best results when comparing performance on more complex datasets such as Clothing-1M.
>
> **Re: Hyper parameter selection methodology:**
>
> This is an interesting point, and the best parameter settings do indeed depend on the dataset characteristics. However, rather than doing a grid search on just one dataset, we have referenced the original publications for guidance on parameter settings.
>
> In section 5 Experimental Setup – Baseline Loss Functions we explain: “Consistent with the original formulation, the GCE hyperparameter q is fixed at 0.7 as is recommended in Zhang & Sabuncu (2018). For the ANL variants, the dataset-specific hyperparameters α, β, and δ, as well as the focal-loss scaling factor γ for ANL-FL, are set exactly as prescribed in Ye et al. (2023)”.
>
> Each of the papers that introduced the respective loss functions performed parameter sweeps to identify the optimal parameters, which we used for all our experiments. For GCE this was a fixed parameter of q = 0.7 for all experiments (same method that was used in the original paper), and for ANL-CE and ANL-FL we use the parameter combinations that for each dataset gave the best results. This ensures that we achieve fair and optimal results for the baseline loss functions, without repeating costly parameters sweeps ourselves. Additionally other loss functions, such as ANL-CE and ANL-FL can have 4 hyperparameters that would need to be tuned for optimal performance, making grid searches computationally expensive, whereas our loss functions use only one hyperparameter, making tuning fairly easy.

---

> > ### Author Response · Authors · 2026-04-10
> > **Response to review, continued**
> >
> > **Re: LNL related work:**
> >
> > We would be happy to add some background on LNL and the methods you listed. However, we would like to clarify that we are not exactly approaching the problem of LNL, but the related problem of label error detection. The goals are subtly different: we seek to identify and propose samples likely to have label errors (even if some are correctly labelled, with trade-offs between precision and recall based on one’s own tolerance in their application) vs. LNL where one seeks a model capable of correctly classifying samples in spite of training on noisy data. An experimental comparison is beyond the scope of this review, as these LNL methods are designed to solve a different problem and do not fit into the framework of detecting mislabeled data. Future work could certainly investigate integrating our methods into LNL pipelines; we appreciate this suggestion. Future work could certainly investigate integrating our methods into LNL pipelines; we appreciate this suggestion.
> >
> > **Re: Clarifying notion of robustness:**
> >
> > We would be happy to clarify the notion of robustness. Presently, in the second paragraph of the introduction, we define robustness: “Existing work on detecting label errors... typically rely on training surrogate models, intended to be robust to overfitting, such that they are well-generalized and not fit to erroneous labels in the training data. Crucially, the effectiveness of these methods depends on the models producing statistically distinguishable predicted probabilities, p(k|x), for erroneously vs. correctly labelled samples”. Then, Section 6.2 Ablation Studies – Training Dynamics, explains how noisy samples have lower loss gradients when using our loss functions, indicating models are ignoring the noisy labels, following our definition of robustness.
> >
> > **Re: Various model architecture experimentation:**
> >
> > Thank you for this suggestion; however, architectures and approaches such as DivideMix and ProMix are meant to solve the LNL problem, rather than the downstream task of label error detection (though, admittedly, there may be overlap in the approaches that are effective in both). Thank you for the suggestion to explore pre-trained architectures, as we may approach that in future work. Nevertheless, our experiments closely mirror experiments from similar papers on robust loss functions and use similar architectures.
> >
> > **Re: Ablation study section label:**
> >
> > Thank you for the suggestion. We chose to divide our results into the simple main study (essentially answering the basic question of whether our proposed loss functions work in general) and then everything else into ablation studies since we felt that the main experiment was already very large scale and fairly comprehensive, and that the smaller-scale experiments included in the Ablation studies section test other aspects of our method.
> >
> > **Re: “Ground truth label” terminology usage:**
> >
> > This is a very fair point you have brought up, and we would be happy to adjust the terminology used in Figure 1 and any other instances throughout our paper to be more precise.

---

### Review · Reviewer_q2a3 · 2026-03-29

**Summary Of Contributions:**

The paper addresses the pervasive issue of label errors in training datasets, which degrades model performance and complicates the detection of such errors. The authors argue that standard loss functions like Cross-Entropy (CE) or Focal Loss (FL) are problematic because they encourage the model to fit to "hard" samples that are frequently just mislabeled outliers. To mitigate the adverse effects of these errors, the authors propose two robust loss functions: Blurry Loss (BL), which de-weights difficult samples, and Piecewise-zero Loss (PZ), which ignores samples with a predicted probability below a certain threshold. A key technical addition is the use of a loss scheduling mechanism that employs a delay parameter, $d$ that allows the model to generalize with standard Cross Entropy before switching to the robust variant. Experimental results on conducted on MNIST, CIFAR-10, CIFAR-100, Food-101, and Clothing-1M datasets and show consistent improvements over the baselines.

**Additional Comments:**

The proposed method use a static delay parameter $d$. I wonder if it is possible to use some kind of annealing mechanism, where in the earlier epochs the loss is more similar to standard CE, and the effect of BL and PZ losses gradually add in as the training progresses?

**Audience:**

Yes

**Audience Explanation:**

The findings of this paper would undoubtedly interest a broad segment of the TMLR audience, especially those focused on supervised learning in domains where mislabeling in training data is common. Since label errors are pervasive even in established benchmarks, the ability to effectively automate error detection has direct implications for reducing data curation costs and improving overall model reliability.

**Claims And Evidence:**

Yes

**Claims Explanation:**

- The proposed BL and PZ losses outperform baselines in nearly all tested datasets in both F1 and Balanced Accuracy scores.
- The experiments include both uniform (artificial) and non-uniform (real-world human) corruption, demonstrating the broad applicability of the method.
- Results on Food-101N (310k images) and Clothing-1M (1M images) confirm the effectiveness of the losses beyond small-scale benchmarks.
- Figure 5, 8 and 9 demonstrate that while parameters $\gamma$ and $c$ need tuning, there is a relatively wide range of values that still outperform baseline methods.

However, I have a few concerns:
1. In the main findings (Section 6.1), the improvements over some baselines (e.g., GCE) are very small, especially on harder datasets like CIFAR-100.
2. Although some larger datasets are used, the number of classes are relatively small. For example, Clothing-1M only has 14 classes. Will the proposed losses work well on datasets with more classes like ImageNet?
3. The idea of the proposed losses is intuitive to prevent overfitting on label noises. However, for the case of feature noises (e.g., huge intra-class deviations, distribution shift over time), it seems that it will hinder learning of those underrepresented examples. In practice, both types of noise can happen. Can you do some experiments to verify that the proposed BL and PZ losses can still consistently perform well under feature noises?

**Requested Changes:**

1.  Can you do some experiments to verify that the proposed BL and PZ losses can still consistently perform well under feature noises?
2. While the authors performed extensive parameter sweeps for their own losses (BL and PZ), they did not perform similar sweeps for the existing state-of-the-art robust loss functions (FL, GCE and ANL). Can you explain how you choose the hyperparameters for the baselines?

---

> ### Author Response · Authors · 2026-04-10
> **Response to review**
>
> Thank you for your detailed review and for recognizing the broad effectiveness of our method. We have responded individually to each of your raised questions and concerns below.
>
> **Re: Number of classes in experiments:**
>
> While running new ImageNet experiments (with artificial corruption) is possible, doing so during the brief review rebuttal timeline is computationally prohibitive. Nonetheless, we feel our current evaluation already demonstrates efficacy well beyond low-class regimes. Specifically, our experiments on CIFAR-100, CIFAR-100N, and Food-101N confirm that our proposed loss functions scale effectively and maintain robust performance on datasets with 100 or more classes, including large-scale, real-world data.
>
> **Re: Experimenting with feature noise:**
>
> This is a valid concern; however, we are specifically trying to address the issue of label errors. One can assume that some degree of feature noise is present in the datasets with which we have experimented, and the results indicate our method performs well.
>
> **Re: Baseline parameter selection:**
>
> In section 5 Experimental Setup – Baseline Loss Functions, we state: “Consistent with the original formulation, the GCE hyperparameter q is fixed at 0.7 as is recommended in Zhang & Sabuncu (2018). For the ANL variants, the dataset-specific hyperparameters α, β, and δ, as well as the focal-loss scaling factor γ for ANL-FL, are set exactly as prescribed in Ye et al. (2023)”. Each of the papers that introduced the respective loss functions performed parameter sweeps to identify the optimal parameters, which we used for all our experiments. For GCE, this was a fixed parameter of q = 0.7 for all experiments (same method that was used in the original paper), and for ANL-CE and ANL-FL we use the parameter combinations that for each dataset gave the best results. This ensures that we achieve fair and optimal results for the baseline loss functions, without repeating costly parameters sweeps ourselves.

---

### Author Response · Authors · 2026-04-10
**Thank you for reviews and commentary**

Dear Reviewers,

Thank you all for your careful review and commentary on our paper. We appreciate all of your questions and suggestions, and will use this feedback to refine our submission. We will separately respond to each of your reviews and address each of your raised points.

We look forward to further discussion and receiving your final reviews.

Thank you

---

### Decision · Action_Editor_HZHT · 2026-05-23

**Recommendation:** Reject

**Additional Comments:**

Reviewer q2a3 is an expert reviewer. The authors are suggested to revise the paper according to his/her comments.

**Audience:**

Yes

**Audience Explanation:**

This paper studies the robust loss function, which is an important topic in machine learning.

**Claims And Evidence:**

No

**Claims Explanation:**

Most of the claims made in the submission are supported by clear evidence. But as one reviewer pointed out, the proposed method is not supported by datasets with more classes and a larger scale. Besides, the proposed method also does not evaluate on some datasets with feature noise.

**Resubmission Of Major Revision:**

The authors may consider submitting a major revision at a later time.